# Intrinsic Capacity to Predict Future Adverse Health Outcomes in Older Adults: A Scoping Review

**DOI:** 10.3390/healthcare11040450

**Published:** 2023-02-04

**Authors:** Jia Zhou, Hui Chang, Minmin Leng, Zhiwen Wang

**Affiliations:** 1School of Nursing, Peking University, Beijing 100191, China; 2School of Nursing, Guizhou Medical University, Guiyang 550025, China

**Keywords:** intrinsic capacity, adverse health outcomes, older adults, scoping review

## Abstract

Objective: Intrinsic capacity is recognized as an important determinant of healthy aging and well-being of older adults; however, relatively little is known about the intrinsic capacity of older adults to predict adverse health outcomes. The study aimed to examine which adverse health outcomes of older adults can be predicted by intrinsic capacity. Methods: The study was conducted using the scoping review methodological framework of Arksey and O’Malley. A systematic literature search of nine electronic databases (i.e., Pubmed, Embase, Cochrane library, Web of science, CINAHL, China National Knowledge Infrastructure, VIP, Wanfang, and the Chinese Biological Medical Literature Database) were performed from the database’s inception to 1 March 2022. Results: Fifteen longitudinal studies were included. A series of adverse health outcomes were assessed, including physical function (*n* = 12), frailty (*n* = 3), falls (*n* = 3), mortality (*n* = 6), quality of life (*n* = 2) and other adverse health outcomes (*n* = 4). Conclusions: Intrinsic capacity could predict some adverse health outcomes of different follow-up times for older adults; however, due to the small number of studies and sample size, more high-quality studies are necessary to explore the longitudinal relationships between intrinsic capacity and adverse health outcomes in the future.

## 1. Introduction

An aging population is one of the most important issues facing society in the development of all countries around the world, and 20% of the world’s population will be over the age of 65 years in 2050 [1]. With the global accelerated increase in life expectancy, more and more attention has been paid to the improvement on the quality of life of older adults [2]. The traditional disease-oriented healthcare approach has proven to be inadequate to meet the complex health and social needs of the rapidly growing number of older adults [3,4]. Consequently, there has been a shift in the focus of aging studies, from a disease-oriented approach to a function-based approach, which aims to establish and maintain the functional ability of older adults.

In 2015, the World Health Organization published the World report on aging and health and proposed the concept of intrinsic capacity. Intrinsic capacity is defined as the composite of all the physical and mental capacities that an individual can utilize at any point in their lifetime [5]. Meanwhile, a functional and person-centered care pathway was developed and the integrated care for older people (ICOPE) screening tool was proposed to identify older adults with a reduced intrinsic capacity by the WHO in 2019, including five domains: cognition, locomotion, sensory (including vision and hearing), vitality and the psychological [6]. Previous research has demonstrated that the five domains of intrinsic capacity are interrelated and inseparable, and none of them can represent the overall level of intrinsic capacity. For example, studies have found that visual impairment and hearing loss can limit the activities of older adults, which in turn affect social and psychological changes, resulting in social isolation, loneliness, and depression [7,8]. Other research results have shown that depression, malnutrition, and vision impairment are closely related to cognitive decline [9,10,11]. Therefore, it is imperative to analyze the integrated intrinsic capacity rather than each domain separately. In addition, intrinsic capacity interacts with relevant environmental characteristics, which has been proven to have a significant impact on an individual’s health status and health trajectory [12]. The concept of intrinsic capacity is of great significance for the development of clinical practice and the formulation of public health strategies, such that it has gradually become a key issue in the field of geriatrics.

It seems crucial to statistically investigate the relationship between the decline in intrinsic capacity and its potential consequences, as it helps to guide us to focus on key older populations as soon as possible, thereby providing a range of interventions to prevent adverse consequences. In particular, applying intrinsic capacity to predict adverse health outcomes may help to capture subtle changes in the early stages of older persons, promoting further personalized health management and promoting personal well-being; however, previous studies were either cross-sectional or only assessed the association between the baseline intrinsic capacity levels and adverse health outcomes [13,14]. It is unclear whether there is a relationship between the intrinsic capacity of older adults at some point in life and the occurrence of adverse health outcomes a few years later. There is solid evidence that each separate domain of intrinsic capacity, such as cognition or depression, can predict individuals’ health status and health trajectories [15,16]; however, since intrinsic capacity is a complete structure, it is equally important to explore the overall predictive power of intrinsic capacity. Recently, a few studies have focused on these domains together as an intrinsic capacity composite concept to predict some health adverse outcomes, including falls, frailty, disability, and the incidence of death in the community or a nursing home [17,18]; however, the follow-up time, outcome type, and predictive effect of intrinsic capacity on adverse health outcomes were unclear.

To our knowledge, there are no published reviews on the impact of adverse health outcomes by intrinsic capacity among older adults. Considering this gap, the purpose of this scoping review was to provide some evidence from longitudinal studies that identify which adverse health outcomes can be predicted by the intrinsic capacity (as a composite structure or its separate domains) of older adults. 

## 2. Methods

Our study was conducted using the scoping review methodological framework of Arksey and O’Malley [19] as well as the guidance for the conduction of systematic scoping reviews developed by Peters et al. [20]. The reporting followed the Preferred Reporting Items for Systematic reviews and Meta-Analyses extension for Scoping Reviews (PRISMA-ScR) guidelines [21]. The PRISMA-ScR checklist is provided in Appendix A.

### 2.1. Searching for Relevant Literature

A literature search was conducted to identify relevant studies that reported the intrinsic capacity in older adults to predict adverse health outcomes. We conducted a three-step search. First, a limited search was conducted in PubMed to develop search strategies tailored to each database. Second, two researchers independently implemented the search strategies in Pubmed, Embase, Cochrane library, Web of science, CINAHL, China National Knowledge Infrastructure (CNKI), VIP, Wanfang, and the Chinese Biological Medical Literature Database (CBM). A combination of MeSH words and free terms was used: “older/aged/elderly/senior/geriatric” and “intrinsic capacity”. The databases were searched for published studies from their inception to 1 March 2022. In addition, all references from the included studies and relevant reviews were also searched. The full search terms and search strategies can be found in Appendix A.

The inclusion criteria were specified as followed: (1) study populations consisting of older adults with a mean age of ≥60; (2) described how intrinsic capacity was measured (either a qualitative or quantitative method); (3) study context including the community, outpatient care, primary care, nursing homes or hospital; (4) a longitudinal study design; and (5) English or Chinese as the publication language. The exclusion criteria were specified as followed: (1) did not detail or clearly report the relationship between intrinsic capacity and adverse health outcomes; (2) unavailability of the full text (e.g., only the title was available); and (3) repeated or overlapped publications.

### 2.2. Selecting Studies

The identified studies from the primary search were downloaded into Endnote X7, a reference management database. Four stages (the identification, screening, eligibility and final inclusion) were implemented for the study selection, followed by a PRISMA screening [22]. After removing duplicate studies, two authors independently screened the titles and abstracts, and then discussed the results to ensure agreement on the included studies. Two authors independently assessed the full text of these studies for eligibility. If there was a disagreement, the authors needed to discuss it to reach a general agreement. If necessary, another author was required to decide whether to include a study.

### 2.3. Charting the Data

All confirmed eligible studies were reviewed as full-text studies using a standardized data extraction form. Two authors independently extracted the following data for each study: the name of the first author, publication year, country/location, data source, study design, total sample, age, sex, follow-up period, setting, outcome measures, and main results. The two independent authors met to ensure the data extracted resulted in the same findings. Similarly, if disagreements about the data extraction of studies occurred between the two authors, a decision was made by the third author. Finally, the data were critically analyzed to look for trends and variances.

### 2.4. Collating, Summarizing, and Reporting the Results

This phase included analyzing the data, reporting the results, and determining the implications of the findings, which was a collaborative process among all the authors. We summarized the results using a narrative descriptive synthesizing approach and presented them in tables and figures. All the study characteristics included in this review are shown in Appendix A. We grouped the studies by the different domains of intrinsic capacity in Appendix A.

## 3. Results

### 3.1. Literature Search

A total of 1439 articles were identified through nine databases in the initial search. The literature search yielded 1044 articles after the removal of duplicates. Following the title and abstract screening, 99 articles were eligible for the next full-text screening. Among the 99 articles, 84 articles were excluded. Finally, 15 studies were found to meet the criteria for inclusion [17,18,23,24,25,26,27,28,29,30,31,32,33,34,35]. The literature screening and selection process is shown in Figure 1.

### 3.2. Study and Participant Characteristics

Twelve studies used data from population-based longitudinal studies, including the English Longitudinal Study on Ageing (ELSA), Sample of Elderly Nursing home Individuals (SENIOR), Longitudinal Assessment of Biomarkers for characterization of early Sarcopenia and predicting frailty and functional decline in community-dwelling Asian older adults Study (GeriLABS), Multidomain Alzheimer Preventive Trial (MAPT), Sarcopenia and Physical Impairment with advancing Age (SarcoPhAge), 10/66 Dementia Research Group (10/66 DRG), pNeumonia and related ConseqUences in nursing home Residents (INCUR study), Yale Precipitating Events Project Study, the MrOS and MsOS (Hong Kong) study, a study of comprehensive geriatric assessment (CGA), and the Beijing Longitudinal Study on Aging II (BLSA II). Three studies collected data from hospital patients or community dwelling recruitment who were not enrolled in a population-based study. Approximately 73% of the studies were conducted in community settings (*n* = 11), and 13% were conducted in nursing homes (*n* = 2) and hospitals (*n* = 2), respectively. The 15 included studies were all longitudinal studies. The time span of these studies ranged from 1 to 21 years, with an average of 4.1 years. Of the 15 studies, China was the most reported on (*n* = 7), followed by Belgium (*n* = 2), the United Kingdom (*n* = 2), France (*n* = 2), Asia (*n* = 1), and America (*n* = 1). Among the 15 included studies, 14 were in English, and 1 was in Chinese. All 15 studies were published from 2019 to 2021, indicating a growing interest in the impact of intrinsic capacity on adverse health outcomes. A total of 36,756 participants were included, with sample sizes ranging from 200 to 17,031 participants. Four studies had a sample size between 1000 to 10,000 participants, while 10 studies had less than 1000 participants. Only one study had a sample size of more than 10,000 participants. The study characteristics are shown in Appendix A.

### 3.3. Measures of Intrinsic Capacity

The five key domains (i.e., locomotion, vitality, cognitive, psychological, and sensory) were proposed by the WHO. According to these five domains, different scholars selected various measurement tools and methods to evaluate the intrinsic capacity of older adults. The measurement tools and methods of studies for the domains of intrinsic capacity are shown in Appendix A.

All 15 studies measured the locomotion domain, and five different measurement methods were identified, including a chair rise/stand (*n* = 12), gait/walking speed (*n* = 11), standing/dynamic balance (*n* = 8), the balance subscale of the Tinetti Performance-Oriented Mobility Assessment (B-POMA) (*n* = 1) and the Tinetti score (*n* = 1). For the vitality domain, nine different measurement methods were identified, including respiratory functioning (*n* = 2), handgrip/grip/muscle strength (*n* = 7), body mass index (BMI) (*n* = 1), abdominal circumference (*n* = 1), mini nutritional assessment (MNA) (*n* = 5), a short-form mini-nutritional assessment (MNA-SF) (*n* = 3), questions about weight loss and appetite loss (*n* = 3), appendicular skeletal muscle mass (ASM) (*n* = 2), and biomarkers (*n* = 1). For the cognition domain, eight different measurement methods were identified, including the time and space orientation plus a word recall (*n* = 1), recall, verbal and letter tests (*n* = 1), mini-mental status examination (MMSE) (*n* = 7), sub-parts of the MMSE (*n* = 2), the eighteen-item Chinese mini-mental status examination (CMMSE) (*n* = 1), the community screening instrument for dementia (CSI-D) (*n* = 1), Hodkinson’s abbreviated mental test (*n* = 1), and the short portable mental status questionnaire (SPMSQ) (*n* = 1). A total of 13 studies measured the sensory domain, and seven different measurement methods were identified, including self-reported vision (*n* = 9), self-reported hearing (*n* = 9), the self-reported Strawbridge questionnaire (*n* = 1), the Jaeger chart (*n* = 1), audioscope (*n* = 1), the Snellen “Tumbling E” chart (*n* = 2), and the Frisby stereo test (*n* = 2). For the psychological domain, 10 different measurement methods were identified, including the eight-item Center for Epidemiological Studies depression scale (CES-D-8) (*n* = 1), self-reported sleep disturbance (*n* = 1), the item “anxiety/depression” of the EuroQol-5D (*n* = 1), two questions of the Center for Epidemiological Studies depression-CES-D (*n* = 1), the ten-item geriatric depression scale (GDS-10) (*n* = 1), the fifteen-item geriatric depression scale (GDS-15) (*n* = 9), the EURO-D depression scale (*n* = 1), the eleven-item Center for Epidemiological Studies depression scale (CES-D-11) (*n* = 1), the Comprehensive Frailty Assessment Instrument (CFAI) (*n* = 1), and the Center for Epidemiologic Studies depression scale (CES-D) (*n* = 1). In addition, Prince et al. [28] considered continence as a domain of intrinsic capacity, including urinary incontinence, faecal incontinence, or both. Incontinence was established from an informant report only, and the capacity was maintained if none of these were reported; however, continence is not currently a primary focus of the ICOPE comprehensive assessment tool, although the guidelines for the assessment and management of incontinence have been prepared by the guideline development group.

### 3.4. Adverse Health Outcomes Predicted by Intrinsic Capacity

Appendix A summarizes 15 studies that examined the relative risk for some adverse health outcomes, predicted by intrinsic capacity. The adverse health outcomes were divided into six categories, namely, the physical function (*n* = 12), frailty (*n* = 3), falls (*n* = 3), mortality (*n* = 6), quality of life (*n* = 2) and other indicators (*n* = 4). Standardized beta-coefficients (*β*) were reported as an outcome measure with positive values indicating a decline in the adverse health outcomes risk in relation to a unit of higher intrinsic capacity performance, and with negative values indicating an increase in the adverse outcomes risk in relation to a unit of lower intrinsic capacity performance. In addition, the odds ratio (OR) or hazard ratio (HR) corresponding to a 95% confidence interval (CI) were calculated to estimate the outcomes. The odds ratio (OR) was most commonly reported, with a value >1 indicating a higher relative risk compared to the reference group, and a value <1 indicating a lower relative risk compared to the reference group. The hazard ratio is an expression of the hazard or chance of events occurring in the treatment arm as a ratio of the hazard of the events occurring in the control arm, and it was reported with a value <1 indicating a risk reduction in the adverse health outcomes with a higher intrinsic capacity at the baseline and a value >1 indicating an increased risk of adverse health outcomes predicted from the intrinsic capacity performance at the baseline. In addition, several studies used the area under the curve (AUC) of the receiver operating characteristic (ROC) to judge the predictive value of the intrinsic capacity to predict adverse health outcomes, and the closer the AUC value was to 1.0, the better the prediction accuracy.

#### 3.4.1. Physical Function

With respect to the studies using measures of physical function as the main outcome, most scholars use activities of daily living (ADLs) and instrumental activities of daily living (IADLs) as measurement variables [36]. ADLs are the skills required to perform daily physical tasks, including dressing, bathing, feeding, moving from a bed to a chair, using the toilet, and maintaining continence. IADLs include more complex activities than basic ADLs, related to the ability to live independently, which constitute doing housework, cooking, shopping, managing money, and taking medication. In the two studies we included, intrinsic capacity predicted the incident loss of ADLs and IADLs limitations both directly and indirectly by ’structural equation modeling (SEM) (*β* = −0.21, *p* < 0.001) [23,31]. In addition, a higher intrinsic capacity was associated with decreased risks of new ADL dependency and new IADL dependency (*β =* 0.14; 95% CI: 0.018–0.29; OR = 0.53, 95% CI: 0.40–0.70; OR = 0.76, 95% CI: 0.61–0.95), where one or more declines in intrinsic capacity strongly and independently predicted the incident dependence (HR = 1.91, 95% CI: 1.69–2.17), and a one-point lower intrinsic capacity (on scale of 0–100) was associated with a 7% increase in the risk for ADL dependency (95% CI: 1.06–1.07) [28,29,30,34]. Moreover, several studies explored the predictive value of the specific domains of intrinsic capacity on IADL or ADL, the results revealed that cognitive decline, limited mobility and depressive symptoms significantly predicted a one-year incident IADL disability (ORs = 2.74–5.48, 95% CI: 1.51–19.88), that malnutrition, limited mobility, visual impairment, and depressive symptoms predicted a three-year and one-year incident of ADL disability (OR = 0.86, 95% CI 0.77–0.96; ORs = 1.80–3.08, 95% CI: 1.06–5.08, respectively), and that a greater value in the vitality domain of intrinsic capacity predicted lower declines in functional status (*β* = 0.04; 95% CI: 0.01–0.07) [17,29,33]. Furthermore, one study found that the overall incidence of ADL disability increased with an impairment in the intrinsic capacity domains (OR = 1.43, 95% CI: 1.14–1.80 for impairment in two domains; OR = 2.32, 95% CI: 1.72–3.11 for impairment in ≥3 domains) at a one-year follow-up [35]. Gonzalez-Bautista et al. [18] also demonstrated that each additional intrinsic capacity conditioned a positive association with a higher risk of incident IADL, and ADL disability, with the risk increased by 27%, and 23% over five years, respectively. Two studies used the AUC to judge the predictive value of intrinsic capacity, and the AUC of the intrinsic capacity for the prediction of ADL and IADL were 0.834, 0.81, and 0.82 [26,30]. All the studies concluded that an impairment in the intrinsic capacity or its domains predicted a significant increase in risk for ADL/IADL disability at follow-up. Meanwhile, the AUC values indicated that intrinsic capacity has a good discriminative power for future prediction. Moreover, Chew et al. [24] showed that a low intrinsic capacity was associated with significant declines in handgrip strength (*β* = −4.1, 95% CI: −5.67–−2.52), gait speed (*β* = −0.08, 95% CI: −0.16–−0.007), and physical function (*β* = −1.2, 95% CI: −2.5–−0.03). 

#### 3.4.2. Frailty

With respect to studies using frailty as the main outcome, Chew et al. [24] conducted a cluster analysis to explore the relationship between frailty and intrinsic capacity in a cohort among community-dwelling older adults, and the results showed that measuring intrinsic capacity may be especially relevant in older adults in the trajectory of frailty. Yu et al. [32] showed that higher scores on intrinsic capacity at the baseline were associated with a lower risk of incident frailty at both follow-ups (i.e., year two—OR = 0.64, 95% CI: 0.59–0.71; year four—OR = 0.64, 95% CI: 0.58–0.71). Another study confirmed that limited mobility (HR = 2.97, 95% CI: 1.85–4.76), depressive symptoms (HR = 2.07, 95% CI: 1.03–4.19), and visual impairment (HR = 1.70, 95% CI: 1.01–2.86) were associated with a higher incidence of frailty over five years [18].

#### 3.4.3. Falls

With respect to studies using falls as the main outcome, two studies evaluated the association between the domains of intrinsic capacity and found that the risk of falling decreased when there was a one-unit increase in the balance performance of the locomotion domain (HR = 0.87, 95% CI: 0.79–0.96), in the nutrition score of the vitality domain (HR = 0.96, 95% CI: 0.93–0.98), and in visual impairment of the sensory domain (OR = 2.85, 95% CI: 1.12–7.21) [17,33]; however, no association was found for intrinsic capacity and repeated falls [17]. Moreover, intrinsic capacity showed a predictive value for the falls (AUC = 0.834, 95% CI: 0.777–0.881) [26].

#### 3.4.4. Mortality

With respect to studies using mortality as the main outcome, a higher intrinsic capacity composite score at admission was associated with decreased risks of six months or a one-year mortality (OR = 0.33, 95% CI: 0.15–0.73; OR = 0.48, 95% CI: 0.31–0.74, respectively) [29,34]. One or more declines in intrinsic capacity (DICs) strongly and independently predicted incident death (HR = 1.66, 95% CI: 1.49–1.85) [28]. A one-unit increase in balance performance and in the nutrition score of intrinsic capacity decreased the probability of death by 12% and 4%, respectively (HR = 0.88; 95% CI: 0.78–0.99; HR = 0.96; 95% CI: 0.93–0.99), and the satisfactory mobility domain, psychological domain and vitality domain of intrinsic capacity appeared to be significantly associated with a reduced mortality risk (HR = 0.45, 95% CI: 0.26–0.79; HR = 0.56, 95% CI: 1.04–3.09; HR = 0.84, 95% CI: 0.70–0.99, respectively) [17,27,29]. Stolz et al. [30] showed that a one-point lower intrinsic capacity value was associated with a 5% increase in mortality (five-year AUC: 0.76, and ten-year AUC: 0.76; 95% CI: 1.04–1.05).

#### 3.4.5. Quality of Life

With respect to studies using quality of life as the main outcome, Yu et al. [33] found that the limited mobility of intrinsic capacity predicted a poor quality of life (OR = 3.03, 95% CI: 1.63–5.66). In another study, quality of life showed a significant decline in low intrinsic capacity (*β* = −0.053, 95% CI: −0.09–−0.02) in a cluster analysis [24].

#### 3.4.6. Other Indicators

With respect to studies using other indicators as the main outcome, Li et al. [25] explored the relationship between intrinsic capacity and a prognosis in older patients with acute coronary syndrome, where a COX regression analysis showed that the intrinsic capacity score was an independent risk factor for the prognosis of older adults (AUC = 0.798, 95% CI: 0.732–0.865). Sánchez-Sánchez et al. [29] found that the cognitive domain of intrinsic capacity was associated with decreased odds of hospitalization (HR = 0.91; 95% CI: 0.84–0.99) and the locomotion domain of intrinsic capacity was inversely associated with the pneumonia incidence (HR 0.84; 95% CI: 0.72–0.98). Stolz et al. [30] found that a one-point lower intrinsic capacity was associated with a 6% increase in the risk for nursing home stays (95% CI: 1.05–1.07). In the study of Yu et al. [33], the results revealed that the cognitive decline and limited mobility of intrinsic capacity significantly predicted emergency department visits during a one-year follow-up (ORs = 2.67–4.22, 95% CI: 1.03–17.24).

## 4. Discussion

The present scoping review aimed to provide some evidence of the intrinsic capacity to predict future adverse health outcomes in older adults. According to our results, the main findings supported the hypothesis that intrinsic capacity at the baseline as a predictor variable has the potential to predict future adverse health outcomes, including physical function impairment, frailty, falls, mortality, a low quality of life, and other indicators.

### 4.1. Summary and Interpretation of Findings

The cognition, locomotion, sensory (including vision and hearing), vitality and psychological domains are the five components of intrinsic capacity; however, some current studies have investigated intrinsic capacity in general, while others have investigated the specific domains. Our results show that impairment in any of these domains predicts a physical function decline in older adults. Most researchers used ADL and IADL as the measurement tools of physiological function, while some studies used the handgrip strength and gait speed. According to a classical theory of aging, intrinsic capacity is a determinant of physical resilience and is also a high-level integrative measure of the physiologic reserve which is the fundamental factor underlying one’s ability to withstand stressors [37]. Due to the continuity of the aging process and disease, individuals with a low physiological reserve or intrinsic capacity impairment could experience poor recovery and may be vulnerable to being disabled once exposed to stressors. Subjects who had impairments in one or more declines in intrinsic capacity had a higher risk of being disabled compared to those without any impairments in the intrinsic capacity domains. This may be explained by the interaction between the intrinsic capacity and relevant environmental characteristics to determine the function of an individual. The worse the intrinsic capacity, the more likely the individual is to be affected by adverse factors in the environment, resulting in a worsened health status of the individual, and a greater risk of disability.

The intrinsic capacity was independently associated with incident frailty, and limited mobility, depressive symptoms, decreased vitality and visual impairment were associated with a higher incidence of frailty. In addition, each additional intrinsic capacity condition demonstrated a positive association with a higher risk of incident frailty. In the study by González-Bautista et al. [18], cognitive decline, decreased vitality, and hearing loss were not significant predictors of frailty, whereas Yu et al. [32] found that vitality was the domain most strongly associated with incident frailty. This is probably due to differences in the study population, the measurement tools, duration of follow-up, statistical methods, and variables included in the analyses, which may differ depending on the purpose and design of a study.

Good performance on the balance of the locomotion domain and nutrition of vitality domain predicted a decrease in the risk of falls. The result was in line with a systematic review and meta-analysis, which found that balance and nutrition were associated with the incidence of falls [38]. Yu et al. [33] found that visual impairment in the sensory domain was predictive of repeated incident falls in a one-year follow-up. The possible explanation for this may be that an impairment of vision is associated with a decrease in ADLs. An avoidance of physical activity in older adults with visual impairment can lead to functional decline and is an important risk factor for falls [39]; however, these findings are inconsistent with Charles et al. [17], who found no association between intrinsic capacity and repeated falls. One of the reasons for such a difference is the duration of follow-up between the two studies, with a longer follow-up being more likely to have a risk of repeated falls.

A decline in intrinsic capacity strongly and independently predicted the mortality risk, whereas a higher intrinsic capacity was associated with a decreased risk in mortality in older adults. For the predictive value of the specific domains of intrinsic capacity, our review found that a satisfactory mobility domain, psychological domain, and vitality domain decreased the risk in mortality. Previous studies have also demonstrated that a good physical function, optimism and good nutritional status could reduce the risk of death mortality, which is consistent with our results [30,31,32,33,34,35,36,37,38,39,40,41,42]. However, we also did not confirm the results of the previous study that revealed an association of mortality risk with cognition [43]. We hypothesize that the duration of the follow-up and the use of other statistical methods could explain this divergence.

A low intrinsic capacity and limited mobility predicted a poor quality of life among community-dwelling older adults. Another systematic review also found that a risk of falls in older adults is associated with reduced postural control, and that falls have a significant impact on older adults’ quality of life [44]. This may be explained by a variety of physiological and psychological mechanisms, such as a decline in muscle strength and mass, aerobic capacity, ADLs, and the psychological function caused by limited mobility [45,46].

Our review also found that the measurement methods and tools of each domain of intrinsic capacity were heterogeneous, and there was little research reporting on the reliability and validity. The most commonly used evaluation methods for each domain of intrinsic capacity were the chair-rise/stand (locomotion), mini nutritional assessment (MNA) (vitality), mini-mental status examination (MMSE) (cognition), questions about vision and hearing (sensory), fifteen-item geriatric depression scale (GDS-15) (psychological), and incontinence (continence). Our results were similar to those of George et al. [47]; however, our scoping review included more original studies. In addition to the continence domain, there were at least five measurement methods for each domain of intrinsic capacity, with a great heterogeneity. Furthermore, even with the same assessment tools, there are differences in the measurements and scoring across studies. For example, several studies used the GDS-15 to measure the psychological domain of intrinsic capacity, but five used the GDS-15 overall scale [25,31,32,34,35], and one used only two items of the GDS-15 [18]. Although the study population was composed of Chinese older adults, the GDS-15 cutoff point was different, and the two studies defined a GDS-15 score of ≥8 as a sign of decline or impairment of psychosocial functioning [25,35], while another study defined it as a score of ≥6 [34]. The reason for this may be that the initial purpose of these studies was to explore the relationship between intrinsic capacity and other variables, rather than scale research. The different measurement methods of the indicators mean that there is no unified scoring standard, which may make it difficult for researchers to select their tools, and this will hinder a comparison between different studies. Therefore, future research should reach a consensus on the measurement and weight of each domain, and explore the reliability and validity of the measurement tools.

### 4.2. Strengths and Limitations

To our best knowledge, this is the first scoping review to examine how intrinsic capacity could predict adverse health outcomes among older adults. Our findings address the knowledge gap and summarize the evidence that could help inform similar studies on the topic; however, several limitations cannot be neglected. A possible limitation of our review is that we only included studies for longitudinal analyses, which reduced the number of studies included in the review. In addition, there is no consensus on how to measure intrinsic capacity, either in terms of the indicator selection, or how it is calculated, weighted, or validated; therefore, we are currently unable to draw fully certain conclusions. Moreover, some studies we included did not report the value of the AUC, resulting in an unclear accuracy of intrinsic capacity to predict adverse health outcomes, and the prediction accuracy needs to be further explored. Furthermore, we did not conduct a meta-analysis but rather qualitatively discussed the results, because the large heterogeneity between the studies made it difficult to directly compare the studies through a quantitative meta-analysis. Finally, although we have fully searched some important electronic databases and clinical registration platforms, the possibility of an insufficient search process still exists. We only included English and Chinese literature, which may also lead to a certain selection bias in our research results.

## 5. Conclusions

In summary, our study indicates that intrinsic capacity could predict physical function, frailty, falls, mortality, quality of life, and other adverse health outcomes at different follow-up times among older adults. Therefore, intrinsic capacity can play an early warning role in the prevention of patients’ adverse health outcomes. Given the significant impact of intrinsic capacity on adverse health outcomes, which can often go unrecognized, future research efforts should focus on the early identification of patients with a reduced intrinsic capacity. However, due to the small number of studies and sample size, more high-quality studies are needed to explore the longitudinal relationships between intrinsic capacity and adverse health outcomes among older adults.

## Figures and Tables

**Figure 1 healthcare-11-00450-f001:**
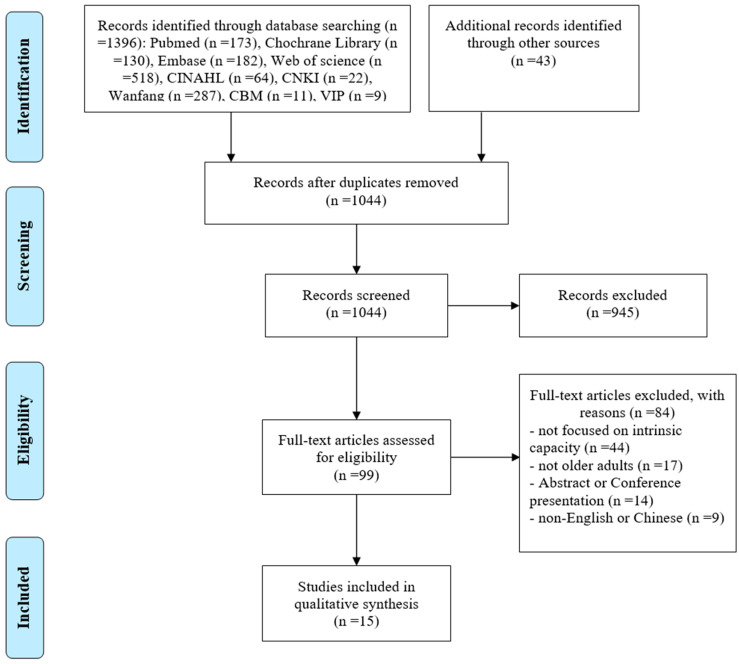
Flow chart of study selection.

## Data Availability

The data presented in this study are available on request from the first author.

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
