# Peer review of "Intrinsic Capacity to Predict Future Adverse Health Outcomes in Older Adults: A Scoping Review"

_healthcare, 2023, doi:10.3390/healthcare11040450_

Round 1

Reviewer 1 Report

The paper investigated the relationship between the intrinsic capacity of older adults and adverse health outcomes through a scoping review, which filled the current research gap. In general, the research is rigorous and the paper is well structured and written. 

Only several minor points need to be addressed.

1. I suggest the authors also provide more specific information on the time span of the reviewed studies in Section 3.1, as this scoping review focuses on longitudinal study, time span is a key research characteristics.

2. Section 4.1: Low intrinsic capacity and cognitive decline, limited mobility, malnutrition, visual impairment and depressive symptoms...I suggest the authors to make it clearer, as in the literature review section, it indicated that cognitive decline, limited mobility and etc. were the components of low intrinsic, though some studies only investigated one dimension of intrinsic capacity.  The current narrative may lead confusion. Better to add explaination, that some reviewed study investigated intrinsic capacity in general, some investigated specific dimensions.

2. Page 2 line 6: ...which has been proven have a significant impact.. -->proven to have.

3. Section 3.4.1 and 3.4.2, reference format chew et al [24]. should be Chew et al. [24], please mind the placement of the dot. 

Reviewer 2 Report

Overall, the review was very inciteful and well-written. I do not have any major comments or minor comments regarding the scope of this review and its content. Well done.

However, some minor comments regarding grammar and context are listed below to help clarify some confusion.

Abstract

1. "... an important determinant of 'healthy' aging ..."

Introduction

Second Paragraph

1. "... can draw on ..." were the authors intending to imply capacities that an individual 'can utilize'?

2. "... which as been proven 'to' have"

3. Last sentence may need to be reworded slightly for clarity.

Third Paragraph

1. "... older persons, 'promoting' further personalized health management ..."

2. Remove "more" from "However, since intrinsic capacity is a complete structure, it is 'more' equally important ..."

3. Remove "And" and change 'focus' to "focused" in the next sentence. Also, the reviewer believes an 'and' should be added before "incidence of death in community and nursing home".

Methods

1. In this section, the reviewers appear to use "authors" and "reviewers" interchangeably. May be best to refer to yourselves as one or the other for consistency in the article.

Searching for relevant literature

1. Last sentence exclusion criteria is numbered from: 1, 2, 4. Please correct "4" to "3".

Selecting studies

1. Extra space before "Two authors independently ..." should be removed.

2. It is mentioned that if a disagreement occurred among the four authors, "the two authors" needed to discuss the issue. It may best to simply state that 'if there was a disagreement, the authors/reviewers needed to discuss ..." 

Results

Literature search

1. Remove "in" in the first sentence "... were identified 'in' through nine databases ..."

Figure 1

1. it is mentioned that n=17 studies were excluded because subjects were not the "elderly." Did the authors mean to write "not older adults" ... as inclusion criteria was older adults greater than or equal to 60 years of age.

3.4. Adverse health outcomes

1. Add 'to' in the sentence: "indicating increase in adverse outcomes risk in relation 'to' a unit lower ..."

3.4.1. Physical function

1. The reviewer may have missed, but is "ADL" and "IADL" defined previously?

2. Did the authors indent to imply a "7% increase in the risk for ADL 'dependency'" in the second sentence.

3. The use of "one-year" and "1-year" is used interchangeably in the article. Please choose one or the other for consistency.

4. "Chew et al ..." should be capitalized in final sentence.

3.4.3. Falls

1. May be able to remove "and the falls" from first sentence

Discussion

4.1. Summary

1. Second last sentence of paragraph may need to be reworded for clarity.

2. The reviewer believes an "and" may need to be inserted after "with incident frailty, ..." in the second paragraph.

3. Third paragraph states "decrease in activities of daily living." Should this be abbreviated as ADL? (Refer to 3.4.1 question 1).

4. 2nd last sentence of paragraph 3 may need some corrections. "However, these findings are inconsistent with Charles et al. [17], which found no ..."

5. Last sentence, may be able to remove "was different"

6. Last sentence, paragraph 5 states: "ability of daily living" was this meant to be "activities of daily living"?

4.2. Strengths and limitations

1. "Our findings address the knowledge gap and summarize the evidence ..."

Reviewer 3 Report

Population aging is one of the most important problems emerging in all countries of the world. The peer-reviewed study was designed to investigate what adverse health outcomes in the elderly could be predicted based on intrinsic capacity. The study was conducted according to the methodological framework of the scoping review, and 15 publications were included in the final analysis.

The main conclusion of the study is that intrinsic ability may predict some adverse health outcomes of different observation times in the elderly. The study indicates that innate ability can predict fitness, frailty, falls, mortality, quality of life and other adverse health outcomes at different follow-up times among older adults. Self-awareness can play an early warning role in preventing adverse health outcomes for patients.

Significant is the authors' conclusion that due to the small number of studies and sample size, more high-quality studies are needed to explore the longitudinal relationships between intrinsic ability and adverse health outcomes among the elderly.
